# Gluten-Free Diet for Fashion or Necessity? Review with New Speculations on Irritable Bowel Syndrome-like Disorders

**DOI:** 10.3390/nu16234236

**Published:** 2024-12-08

**Authors:** Raffaele Borghini, Alessia Spagnuolo, Giuseppe Donato, Giovanni Borghini

**Affiliations:** 1Stella Maris S.T.P.—Gastroenterology Unit, Via Giuseppe Prina, 8, 00139 Rome, Italy; 2Medical Oncology, Department of Radiology, Oncology and Pathology, Sapienza University of Rome, 00155 Rome, Italy; alessia.spagnuolo@uniroma1.it; 3Gastroenterology Unit, Department of Translational and Precision Medicine, Sapienza University of Rome, 00155 Rome, Italy; giuseppe.donato@uniroma1.it; 4Stella Maris S.T.P.—Food and Human Nutrition Unit, 00139 Rome, Italy; giovanni.borghini@tiscali.it

**Keywords:** celiac disease, gluten-free diet, irritable bowel syndrome (IBS), IBS-like disorders, non-celiac gluten sensitivity (NCGS), nickel allergy, intolerance to lactose, FODMAPs, histamine intolerance, diet

## Abstract

Nowadays, the gluten-free diet (GFD) has become much more than the dietary treatment for celiac disease. Due to its presumed beneficial effects even in non-celiac subjects, it has become a new fashion statement and it is promoted by some healthcare professionals, social media and marketing strategists. On the other hand, regardless of a proper medical indication, a GFD may present side effects, such as poor palatability, high costs and socio-psychological adversities. Moreover, it can be an obstacle to correct clinical practice and may induce nutritional deficiency due to a low-quality diet. In addition, a GFD can trigger or exacerbate many irritable bowel syndrome (IBS)-like disorders in predisposed subjects: reactivity to dietary nickel, the increased consumption of FODMAP-rich foods and histamine intolerance seem to frequently play a relevant role. The possible intersections between high-risk foods in these categories of patients, as well as the possible overlaps among IBS-like disorders during GFD, are described. In conclusion, it is advisable to undergo a careful clinical evaluation by a gastroenterologist and a nutritionist (in some cases, also a psychotherapist) before starting and during a GFD, because both benefits and risks are possible. It is also important to take into account IBS-like disorders that can be exacerbated by a GFD and that are still underestimated today.

## 1. Introduction

An appropriate gluten-free diet (GFD) is currently the only known therapy for the treatment of celiac disease (CD) together with its intestinal and extra-intestinal signs and symptoms; in this condition, this kind of treatment can bring multiple benefits. However, its use is currently also spreading to other clinical contexts, even without medical control or indication. In this regard, new evidence reveals that the GFD hides a possible threat, which gastroenterologists, nutritionists, celiac patients and those who adopt this dietary style should be aware of.

## 2. Celiac Disease and Gluten-Related Disorders

Nowadays, the pool of gluten-related pathologies has been expanded beyond CD and mainly includes non-celiac gluten sensitivity (NCGS) and Wheat Allergy (WA).

### 2.1. Celiac Disease (CD)

CD is an autoimmune disease caused by the ingestion of gluten and affects approximately 1% of the population. Specifically, it affects a genetically predisposed population, HLA DQ2+ (>90%) and/or DQ8+ (5%) [1].

Despite having an intestinal trigger, CD can also show systemic effects and affect multiple organs, thus causing both intestinal (bloating and abdominal pain, diarrhea, constipation, vomiting, etc.) and extra-intestinal symptoms (dermatitis, headache, muscle and joint pain, etc.).

Its diagnosis is based on the histology of duodenal biopsies performed during a gluten-containing free diet, characterized by villous atrophy, crypt hyperplasia and intraepithelial lymphocytosis. In most cases, positive results for Anti-Endomysial antibodies (EMA), anti-tissue transglutaminase (anti-tTG) and Anti-Deamidated Gliadin Peptide (AGA DGP) serological antibodies (IgA and IgG) can be found [2]. Recently, a no-biopsy diagnosis in children has been introduced, to be used when the IgA anti-tTG antibodies are >10 ULN, associated with EMA-positive results [3].

In recent decades, other diagnostic methods have also been described and proposed, especially for doubtful or borderline cases, such as the search for specific antibodies in supernatants of cultured human duodenal biopsies (“organ culture system”) and mucosal anti-transglutaminase IgA deposits in duodenal biopsies (“immunohistochemistry method”) [4].

Currently, a strict and life-long GFD is the only known therapy capable of inducing and maintaining the remission of CD, although new possible approaches have been proposed, such as the use of deglutinated wheat, whose reliability is still to be fully clarified [5].

The objectives of GFD in CD treatment are mainly (1) the verification of effective and efficient dietary adherence; (2) the assessment of CD serological and histological remission; and (3) screening for complications. To date, there is still no standardized algorithm or timing to evaluate correct adherence to GFD, but the finding of persistent villous atrophy on duodenal biopsy during a GFD, as well as the persistence of positive EMA and anti-tTG serological results after a significant period of time, can lead to a diagnosis of non-responsive CD and, in more severe cases, refractory CD. Unfortunately, clinical symptomatic data alone are not sufficient to verify correct adherence to GFD, since there are cases of asymptomatic or pauci-symptomatic CD patients. Moreover, additional factors often intervene during a GFD to exacerbate pre-existing symptoms or even cause new ones to arise, as will be described in detail later [6].

Correct adherence to a GFD can be conditioned by numerous factors, especially those related to gluten contamination. In this regard, the Food and Drug Administration (FDA) has designated a gluten limit of 20 ppm that must not be surpassed to guarantee tolerance. Perhaps the most relevant role in GFD contamination is played by cross-contact, which may happen at any stage (production, transport, sale and consumption of food), away from home (restaurants, canteens, etc.), and in the domestic environment (“kitchen cross-contact”). In addition, there is still an open debate about the potential contamination risk of some foods, such as oats: their consumption is very controversial, and even today, there are no univocal indications on this matter, so only cautious use can be suggested, with strict follow-up to assess any possible adverse effects. Last but not least, poor quality of life and/or negative psychosocial aspects can also affect correct adherence to the GFD. In some cases, a GFD can negatively affect social domain areas (e.g., dining out, travel, work, partner burden), may induce states of anxiety/depression and can even result in voluntary contamination by the patient themselves [7].

#### Positive Aspects of GFD in Celiac Disease

There are many potential benefits of a GFD in CD, especially the following:-Malabsorption syndrome treatment: One of the most significant changes concerns nutritional status, with frequent improvement in microcytic iron deficiency anemia, serum proteins and nutritional indices, such as normal values of glucose and lipid profiles in those who had low serum levels before starting a GFD. Even the absorption of vitamins (especially vitamin D, folate and B12) and other minerals (copper, zinc) can increase if adequately implemented, contributing to general well-being. Recovery from malnutrition occurs not only because of the resolution of malabsorption, but also as a result of increased oral intake due to the progressive resolution of intestinal symptoms. In addition, the energy expenditure for intestinal mucosal regeneration is increasingly reduced as the GFD becomes more effective [7].-Improvement in body composition and strength: A GFD can also lead to improvements in body composition variables, such as fat mass, body mass index (BMI) and fat-free mass, although supporting resistance training seem to be essential [8].-Improvement in psychological health: Over time, it has also been highlighted that a GFD in CD patients can lead to a positive change in well-being and can induce a significant reduction in depressive states. When socio-demographic features are analyzed, the most susceptible categories of people that seem to suffer the most from a psychological point of view are women, the elderly and the poorly educated [9].-Reduction in the gluten-dependent inflammatory state: There are several studies that have shown a reduction in circulating proinflammatory cytokines during a GFD in CD patients, such as interferon-γ, interleukin (IL)–1β, tumor necrosis factor–α, IL-6 and IL-8, and also Th-2 cytokines such as IL-4 and IL-10 [10]. Furthermore, a GFD in CD subjects appears to reduce the visceral and subcutaneous inflammatory signal coming from immune cells related to adipose tissue infiltration [11].-Improvement in intestinal/extra-intestinal symptoms: Evidence suggests that a correct GFD can cause significant improvements in the intestinal symptoms of typical CD; these are mainly linked to malabsorption and include diarrhea, abdominal pain and swelling, dyspepsia, vomiting, chronic constipation, growth retardation in children, anorexia and weight loss. Similarly, there are numerous extra-intestinal manifestations of atypical CD patients that may improve with a proper GFD, such as dental enamel hypoplasia; recurrent oral aphthae; hypostatism; hepatitis and elevated transaminase levels; arthritis and osteoporosis; dermatitis herpetiformis; infertility; headache; ataxia; epilepsy; and other neurological manifestations [12].-Possible shield against comorbidities: Even though there is no consensus in the scientific literature, proper GFD and CD treatment are supposed to avoid or mitigate possible comorbidities and complications, such as other autoimmune diseases (e.g., autoimmune thyroiditis, systemic lupus erythematosus, type 1 diabetes, hepatitis, vasculitis, arthritis, Sjögren’s syndrome) [2]. The hypothetical protective role of the GFD in cancer development in CD patients is not clear either. However, some studies have shown that late diagnosis, the type of cancer and the type of CD can influence it. The most common CD-associated cancers are non-Hodgkin’s lymphoma, specifically enteropathy-associated T-cell lymphoma (EATL), followed by Hodgkin’s lymphoma, colon carcinoma, adenocarcinoma of the small intestine and thyroid cancer [13].

### 2.2. Non-Celiac Gluten Sensitivity (NCGS)

NCGS is a clinical condition characterized by intestinal and extra-intestinal gluten-dependent symptoms, although with a total absence of CD serological or histological features. Moreover, Wheat Allergy (WA) should be excluded for a correct differential diagnosis.

NCGS’s estimated prevalence is 6%, although according to some authors, the percentage can vary from 0.6% to even 13% of the general population.

Similarly to CD, NCGS may have intestinal manifestations (e.g., diarrhea, abdominal pain, abdominal bloating) as well as extra-intestinal symptoms (e.g., headache, foggy mind, attention deficit/hyperactivity disorder, ataxia, recurrent oral ulcers, psoriasis) [14].

Specifically, serological IgG anti-gliadin (AGA) positivity was observed in >50% of NCGS subjects, whereas HLA DQ2 and/or DQ8 positivity was observed in less than 50% of cases. A duodenal biopsy of subjects on a gluten-containing diet showed no alterations or simply non-specific intraepithelial lymphocytosis [6]. Recent ultrastructural investigations using transmission electron microscopy of the duodenal biopsies of NCGS patients showed significant shortening of intestinal microvilli, as well as a significant increase in inter-villous spaces compared to controls, but of an intermediate degree compared to active CD. Regarding junctional complexes, tight junctions did not show significant differences even compared to controls, whereas adherens junctions appeared significantly more dilated and the distance between desmosomes was greater compared to controls. However, although very useful, these are preliminary and non-specific data, as they can also be detected in other pathological conditions [15].

The diagnosis of NCGS is currently based on a double-blind placebo-controlled gluten challenge proposed by the Salerno experts, which involves a GFD of at least 6 weeks and the alternating inclusion and exclusion of gluten in the diet for 3 weeks [16]. However, limitations and contradictory results of this diagnostic protocol have emerged; in fact, the protocol seems to be very long, complex and difficult to apply in real-life clinical practice, as it requires close follow-up by the specialist and considerable compliance by patients, who often refuse to reintroduce gluten into their diet and prefer to self-diagnose NCGS and start a GFD [14].

Since specific diagnostic biomarkers for NCGS are missing, other methods have also been proposed, such as a gluten oral mucosa patch test, ultrastructural research on duodenal alterations including zonulin, ALCAT test, micro-RNA, incRNA and many cytokines [15,17,18].

As for CD, NCGS patients on a GFD also require specific nutritional assessment and management. In particular, nutritional screening is necessary, with the categorization of nutritional status and potential risks (under- or hyper-nourishment, macro- and micronutrient imbalance). Also, in this case, a multidisciplinary team composed of a gastroenterologist and nutritionist is necessary to verify correct adherence to the GFD and its nutritional adequacy [19].

### 2.3. Wheat Allergy (WA)

WA is a pathological entity that is characterized by a classic IgE-mediated or type 1 allergic reaction to wheat. It affects 0.1% of the population and can have intestinal symptoms similar to CD (diarrhea, abdominal pain and swelling), as well as skin manifestations (dermatitis). Other clinical features besides food allergy may include respiratory allergy, urticaria and even wheat-dependent exercise-induced anaphylaxis (WDEIA).

Since it is caused by the ingestion of wheat—but not specifically gluten—it is not always easy to make a differential diagnosis with CD. Also, in this case, as in NCGS, there is no specific correlation with HLA DQ2 and/or DQ8, nor is there identifiable positivity for serological EMA, anti-tTG or AGA. Histological examination of duodenal biopsies does not show significant alterations, with no signs of villar atrophy. The correct diagnosis of WA is based on careful anamnesis, a skin prick test, specific serological IgE, Basotest and ral provocation test. For a correct differential diagnosis, it is also advisable to consider specific IgE-mediated reactions to other cereals such as barley, rye and oats, as well as gluten.

Therapy for WA is based on a wheat-free diet, and antihistamines may reduce signs and symptoms of minor WA, to relieve discomfort. Moreover, injectable doses of epinephrine are an emergency treatment for anaphylaxis in case of a severe reaction to wheat [20].

### 2.4. GFD for Other Conditions

Some authors have proposed a GFD in other different pathological conditions, even though the results have appeared unclear [21]:-Irritable bowel syndrome (IBS): It has long been observed that some IBS patients benefit from a GFD, and studies have been conducted on this topic. However, although a GFD in some cases has been associated with a reduced risk of experiencing overall symptoms, recent scientific evidence and meta-analysis have shown that a GFD alone is not sufficient to provide statistically significant benefits and therefore should not be recommended in routine clinical practice except in a very limited subgroup of patients [22].-Neurological and psychiatric diseases: A reduction in or the elimination of gluten from the diet has shown interesting beneficial results with symptoms such as depression, anxiety or even cognition deficiency, and to a lesser extent for schizophrenia and autism spectrum disorder [23]. The connection between diet and these types of pathologies probably resides in a network that includes gut flora and immune system dysregulation, oxidative stress, nutrient deficiencies and the physicochemical and nutritional effects of foods [24]. However, more studies on the subject are necessary to obtain more reliable conclusions and indications.-Psoriasis: There seems to be a close connection between psoriasis and intestinal pathologies, in particular, those related to serological positivity for AGA. Although not much is known about the etiopathogenetic bases, a GFD seem to induce some improvement in psoriatic manifestations [25].-Endometriosis: Many endometriosis patients follow a GFD as advised by social media and patient forums, but it has been highlighted that, unless CD or NCGS is diagnosed, it has numerous adverse effects and is therefore not recommended for the management of endometriosis-related symptoms [26].

## 3. Gluten-Free Diet: A Real Necessity or Purely a Fashion Statement

In recent years, there has been a dramatic increase in the consumption of gluten-free products, and this phenomenon involves more and more subjects who are asymptomatic or have symptoms that have not been proven to clearly arise from gluten ingestion. The reasons are often found to include a desire for a healthier lifestyle, as well as weight loss, a reduction in cardiovascular risk, and the improvement of physical performance, although sometimes, these targets seem to be merely attributable to a reduction in processed foods and an increase in naturally gluten-free whole foods (e.g., fruits, vegetables, gluten-free grains) instead of to gluten avoidance itself [27]

A predominant role in this emerging habit can be surely attributed to aggressive commercial advertising and the increase in nutritionists indiscriminately promoting a GFD, although scientific data on the nutritional benefits of a GFD are conflicting. Moreover, the GFD is strongly recommended by social media influencers and sports/cinema celebrities, often free from any nutritional or gastroenterological surveillance. Finally, the GFD may involve not only a single individual but also the whole family unit, including the children, for both ideological and practical reasons [28].

### Negative Aspects of GFD

The possible “dark sides” of a GFD, especially in the absence of a real medical need, have been increasingly highlighted and can be summarized as follows (Figure 1):-Poor palatability: In particular, gluten-free baked goods have a much less elastic and much drier dough than their gluten-containing counterparts [29].-High costs: Gluten-free products are reported to be up to five times more costly than standard gluten-containing products [30].-Social and psychological adversities: A GFD inevitably affects the life not only of the single individual, but also of all the people around them, from family to friends, from the work environment to all places involved in the food system. Often, food choices, the places where food is prepared or consumed can end up limiting people’s social and working life and can represent a burden that cannot always be overcome [31]. From a psychological point of view, a GFD can cause depression and poor quality of life [32], as it can be the cause or a consequence of orthorexia nervosa as well [33].-Obstacle to correct clinical practice: An indiscriminate GFD negatively affects a correct diagnostic pathway towards CD, NCGS or WA, especially if gluten-related signs and symptoms are present. Once the GFD is undertaken, it is necessary to reintroduce appropriate quantities of gluten into the diet for an adequate period of time, which is not always feasible due to the reappearance/exacerbation of gluten-dependent symptoms and psychological resistance [27].-Risk of nutritional deficiency and low-quality diet: The consumption of gluten-free products may lead to a very selective diet, especially in the absence of expert professional guides. Moreover, a GFD may include processed and packaged foods, at the expense of naturally gluten-free whole foods. Macro- and micronutrient imbalance associated with a GFD can lead to an increase in saturated fats and lipids, simple carbohydrates and sodium, as well as a decrease in proteins, complex carbohydrates and fibers, zinc, folate, iron, calcium, vitamin B12 and vitamin D [34].-Exacerbation or onset of other comorbidities: The GFD has already been related to heavy metal exposure. For example, an increase in alimentary nickel during a GFD has been demonstrated, and this can be responsible of the onset/worsening of gastrointestinal and extra-intestinal symptoms in nickel-sensitive patients. However, the accumulation of other potentially harmful agents during a GFD cannot be excluded [35].

## 4. Gluten-Free Diet: Possible Cause of IBS-like Disorders and Symptoms

There is a growing awareness that a GFD can lead to increased consumption of foods rich in fermentable oligo-, di- and monosaccharides and polyols (FODMAPs) and nickel; both of these conditions can cause or worsen IBS-like intestinal symptoms, as well as extra-intestinal manifestations, in specific categories of patients. Histamine intolerance should also be included in this risk group.

### 4.1. FODMAPs

FODMAPs are small osmotically active molecules that include lactose, fructose, fructans, galacto-oligosaccharides and polyols (sorbitol, mannitol, xylitol, and maltitol). They are poorly absorbed because of the absence of luminal enzymes capable of hydrolyzing the glycosidic bonds or absence/low activity of brush border enzymes (e.g., lactase), the presence of low-capacity epithelial transporters (e.g., fructose, glucose transporter 2 [GLUT-2], glucose transporter 5 [GLUT-5]) or because they are simply too large for diffusion (e.g., polyols). Thus, FODMAPs can cause gastrointestinal symptoms due to excessive fluid and gas accumulation, and the chain length of the carbohydrate determines the fermentation rate [36].

A GFD—with or without a CD diagnosis—may lead to a FODMAP overload (e.g., from legumes, amaranth or many vegetables and fruits) and, over time, exacerbate IBS-like symptoms. In this regard, recent data showed that a low-FODMAP diet can improve intestinal symptoms in both IBS and CD patients; thus, it is often recommended by gastroenterologists and nutritionists [37,38,39]. On the other hand, eliminating high-FODMAP foods during a GFD seems to only partially resolve actual risks and disorders.

### 4.2. Nickel Allergic Contact Mucositis

Reactivity to dietary nickel is a pathological condition that is increasingly studied given its high estimated prevalence (even > 30%). It has often been called “Allergic Contact Mucositis (ACD)” to be differentiated from “Allergic Contact Dermatitis (ACD)”, although they are two sides of the same coin (“Systemic Nickel Allergy Syndrome (SNAS)”) (Figure 2). Currently, the gold standard for its diagnosis is the nickel skin patch test, although a similar version has also been proposed to be performed on the oral mucosa, probably capable of avoiding false negative results.

Therapy for this condition consists of a reduced intake of Ni-containing food, the use of natural clinoptilolite zeolite as ion exchanger, iron supplementation (nickel and iron compete for the same intestinal transporter, Divalent Metal Transporter 1—DMT1) and ascorbic acid supplementation to reduce nickel concentration in plasma and increase antioxidant defense [40].

A dietary nickel overload can be observed in CD patients during a proper GFD and even on a low-FODMAP diet (e.g., due to overconsumption of buckwheat, millet, brown rice, whole quinoa, sorghum, corn or potatoes), causing the persistence, exacerbation or de novo appearance of IBS-like intestinal symptoms and extra-intestinal manifestations. In this regard, a low-nickel GFD has already shown brilliant results in the management of IBS-like symptoms in this category of patients [35].

### 4.3. Histamine Intolerance

Histamine intolerance or enteral histaminosis or sensitivity to dietary histamine is a disorder that occurs due to reduced histamine degradation capacity in the intestine caused by impaired Diamine Oxidase (DAO) activity; the consequent histamine accumulation in plasma seems to be related to many adverse effects, included IBS-like gastrointestinal ones. The influencing factors in histamine intolerance are summarized in Figure 3 [41].

Although there are still no certain data, histamine intolerance’s estimated prevalence is 1–3%, a percentage that may even increase with the spread of knowledge and diagnostic methods. Its may be diagnosed when there is clinical evidence of ≥2 symptoms in <4 h after histamine-rich food intake and improvement/remission after a low-histamine diet. It is confirmed by DAO activity testing in blood samples or intestinal biopsy, as well as by genetic and metabolic markers.

A low-histamine diet, direct DAO supplementation and H1R antihistamines may be useful in controlling symptoms, as may the administration of DAO cofactors (ascorbic acid, copper, vitamin B6) [42,43]. Also, in this case, a GFD can dangerously favor histamine-rich or histamine-releasing foods, causing worsening of symptoms in predisposed subjects; this happens, for example, in cases of high consumption of legumes, dried fruit, nuts, mushrooms and some types of fruit and vegetables (e.g., aubergine, spinach, tomato, avocado, bananas, strawberries) [44].

Figure 4 summarizes the main foods high in FODMAPs, nickel and histamine, whose consumption can significantly increase during a GFD. Note that there is considerable overlap amongst these categories, further increasing the risk of causing or worsening IBS-like symptoms.

## 5. Discussion

A whole new range of applications have been established for the GFD. Initially, it only used for CD and its treatment; its uses and benefits, as well as its availability, have improved more and more, and it has become a precious resource for CD patients and healthcare professionals. Subsequently, its use has also expanded under the pressure of the food industry and trends, to the point of reaching therapeutic purposes even beyond CD, and some of these are still being studied, with controversial results.

On the other hand, numerous studies have begun to highlight the possible dark sides of this type of diet, especially if detached from its original purpose. Dietary interventions can lead to weight gain and negative metabolic changes such as dyslipidemia, fatty liver disease and insulin resistance, resulting in increased cardiovascular risk [45]. They can also affect the gastrointestinal tract and quality of life; this is also what happens when a GFD is started for medical or other reasons. In such radical conditions, strict clinical vigilance is necessary, but a nutritionist alone may not be enough. In fact, a gastroenterologist can recognize possible organic/inflammatory comorbidities or IBS-like disorders, which may already be present at the time of the dietary intervention, or which may be induced/exacerbated by the special diet itself. Additionally, the involvement of a psychotherapist can be of great help in managing these increasingly frequent conditions.

As can be seen in Figure 5, there is a complex network of factors that can influence IBS-like disorders and symptoms, and they can range across the field of integrative medicine. Nowadays, it is necessary to consider not only possible comorbidities, but also nutritional interventions, anatomical factors, lifestyles and even the bio-psycho-social model of patients, especially when handling their diet and the related consequences.

For example, as described above, a GFD can trigger an allergic reaction to nickel or an overload of high-FODMAP foods, or it can exacerbate a histamine intolerance.

To make things worse, a severe dolichocolon with no macroscopic organic or inflammatory alterations may also be responsible for an intestinal transit time even ×3.5 times longer than normal, with all the discomfort that this may represent for the patient in terms of IBS-like symptoms, especially after diet changes [46].

In addition, wrong eating habits, scarce fluid intake and a high dietary consumption of fats, coffee, alcohol and spicy foods can worsen IBS-like symptoms and make any diet more difficult to follow [47,48,49].

Last, but not least, psychological and social factors can interfere with the communication between the central nervous system (CNS) and enteric nervous system (ENS). Examples of influential factors are neuroticism, hypochondrial beliefs, major depression, dysthymic disorders, generalized anxiety, abuse history, the divorce or death of a parent and social learning (modeling). This bio-psycho-social model is believed to be involved in the onset of IBS and can influence both the treatment (diet included) and outcome [50].

## 6. Conclusions

In conclusion, major changes in dietary regimen such as undertaking a GFD may have benefits, although sometimes they are only temporary and/or partial. On the other hand, there can also be many negative effects, tipping the balance towards other IBS-like disorders (e.g., Nickel Allergic Contact Mucositis, FODMAPs and histamine intolerance) or stressors. A clinical approach to patients on a GFD who develop digestive and extra-intestinal symptoms is suggested by the algorithm in Figure 6.

Thus, new insights and perspectives are now available that indicate that some IBS-like disorders are still probably underestimated and can often overlap, making the clinical picture more complex to interpret. It is therefore advisable that a unit of integrated specialists composed of a gastroenterologist, a nutritionist and even a psychotherapist monitors such dietary conditions, in order to verify the appropriateness of the GFD and safeguard all its inherent aspects.

## Figures and Tables

**Figure 1 nutrients-16-04236-f001:**
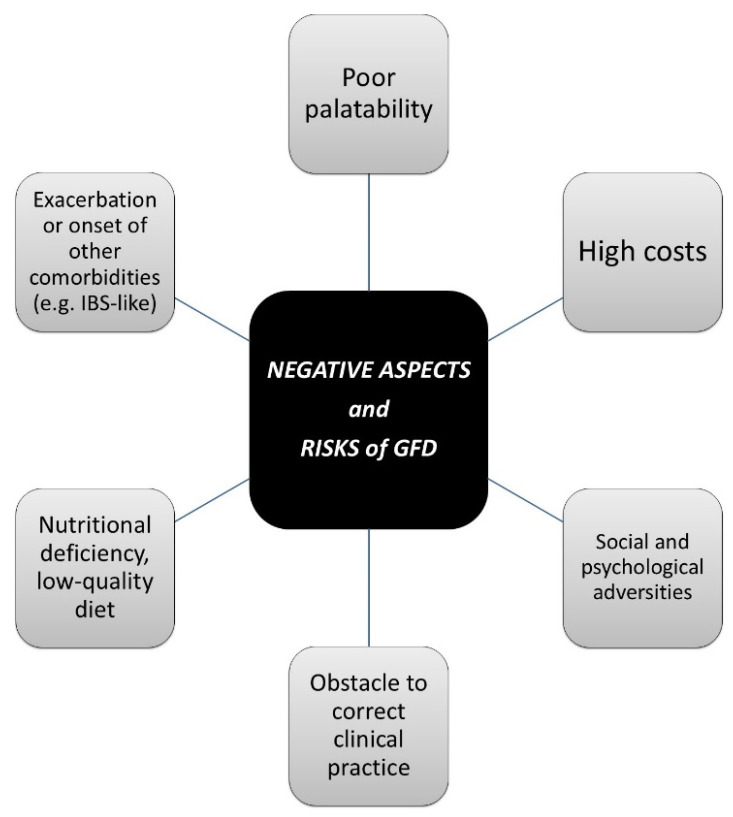
Negative aspects and risks of GFD. Legend: GFD, gluten-free diet; IBS, irritable bowel syndrome.

**Figure 2 nutrients-16-04236-f002:**
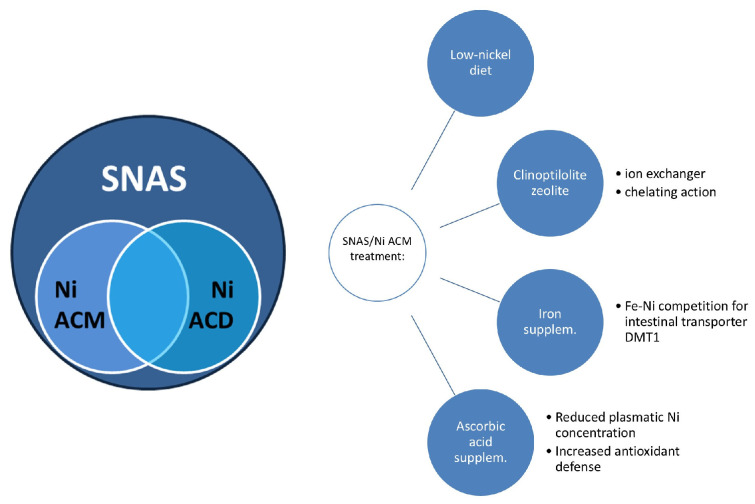
Nickel allergy and SNAS treatment. Nickel Allergic Contact Mucositis and Nickel Allergic Contact Dermatitis can be considered two sides of the same coin (“Systemic Nickel Allergy Syndrome”). The main treatments are reported. Legend: Ni ACM, Nickel Allergic Contact Mucositis; Ni ACD, Allergic Contact Dermatitis; SNAS, Systemic Nickel Allergy Syndrome.

**Figure 3 nutrients-16-04236-f003:**
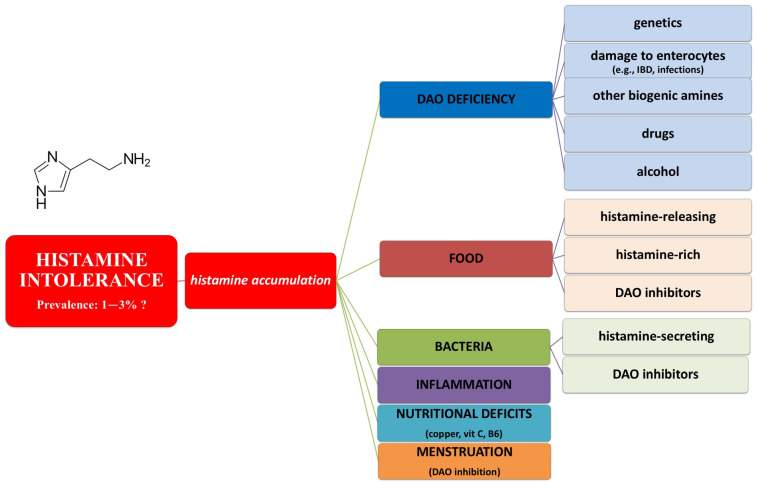
Etiopathogenesis of histamine intolerance. The flow-chart shows the main factors involved in histamine accumulation in histamine intolerance. Both primitive (congenital) and secondary (acquired) factors are reported. Legend: DAO, Diamine Oxidase; IBD, inflammatory bowel disease.

**Figure 4 nutrients-16-04236-f004:**
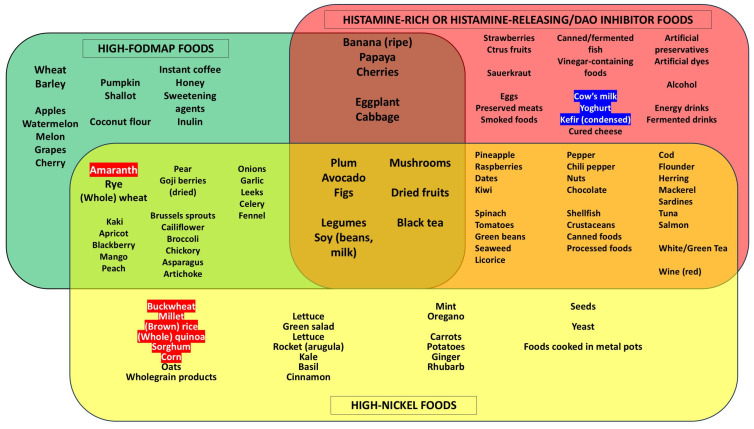
Foods with high nickel content and their possible overlap with foods rich in FODMAPs and histamine-rich/histamine-releasing/DAO inhibitor foods. Note the risk overlap between pairs of categories and even among all three categories, which can worsen during a gluten-free diet or uncontrolled nutritional interventions. The naturally gluten-free cereals are marked in red: note how they are mainly concentrated among foods with a high nickel content; therefore, they can trigger or aggravate IBS-like symptoms in predisposed subjects. The foods containing lactose are marked in blue, which can further aggravate IBS-like manifestations. Legend: DAO, Diamine Oxidase; FODMAPs, fermentable oligo-, di-, and monosaccharides and polyols.

**Figure 5 nutrients-16-04236-f005:**
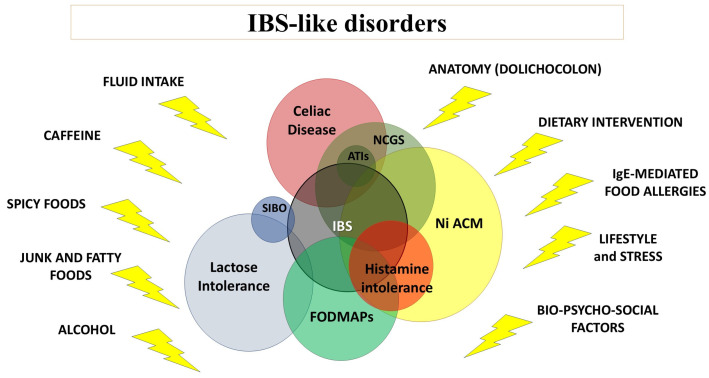
Overlapping IBS-like disorders and their possible modulating factors/comorbidities. The graph highlights the possible clinical overlap among IBS and other IBS-like disorders. The yellow lightning symbols represent the external factors that can influence their manifestations. Modified by Borghini R et al. [40]. Legend: IBS, irritable bowel syndrome; FODMAPs, fermentable oligosaccharides, disaccharides, monosaccharides and polyols; SIBO, Small Intestinal Bacterial Overgrowth; NCGS, non-celiac gluten sensitivity; ATIs, α-Amylase/Trypsin Inhibitors; Ni ACM, Nickel Allergic Contact Mucositis.

**Figure 6 nutrients-16-04236-f006:**
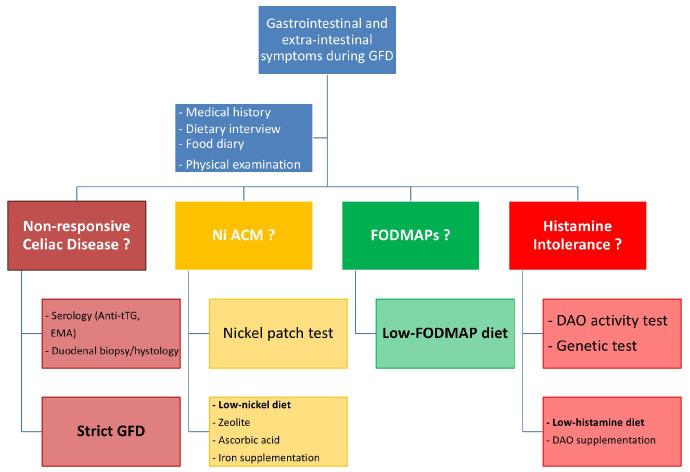
The clinical approach to patients on a GFD who develop symptoms. This algorithm proposes a possible diagnostic and therapeutic path for symptomatic patients on a GFD. Legend: anti-tTG, anti-tissue transglutaminase antibodies; DAO, Diamine Oxidase; EMA, Anti-Endomysial antibodies; FODMAPs, fermentable oligosaccharides, disaccharides, monosaccharides and polyols; GFD, gluten-free diet; Ni ACM, Nickel Allergic Contact Mucositis.

## Data Availability

Not applicable.

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
