# Peer review of "Gluten-Free Diet for Fashion or Necessity? Review with New Speculations on Irritable Bowel Syndrome-like Disorders"

_nutrients, 2024, doi:10.3390/nu16234236_

Round 1

Reviewer 1 Report

Comments and Suggestions for Authors

 1. It is recommended to spell out the full term for IBS-like disorders in the article title and abstract.

2. The title reads: "Gluten-Free Diet: Fashion or Necessity - A Double-Edged Sword? A Brief Review on New Speculations about IBS-like Disorders." What are these new speculations? The author's new insights are not evident.

3. I suggest adding a paragraph in "2. Celiac Disease and Gluten-Related Disorders" that provides an overview of the necessity of a gluten-free diet in these conditions.

4. Figure 1 is included in the document but lacks mention in the main text.

5. Figure 5 shows potential clinical overlaps between IBS and other IBS-like disorders. Yet, ATIs, FODMAPs, and SIBO are listed as factors rather than diseases. Please correct Figure 5 accordingly.

6. It is advised to avoid using single-sentence paragraphs, such as those found in lines 188-189.

Author Response

REVIEWER 1 COMMENTS and ANSWERS

  1. It is recommended to spell out the full term for IBS-like disorders in the article title and abstract.

Thank you for your suggestion. Appropriate changes have been made.

  1. The title reads: "Gluten-Free Diet: Fashion or Necessity - A Double-Edged Sword? A Brief Review on New Speculations about IBS-like Disorders." What are these new speculations? The author's new insights are not evident.

Thanks for the comment. The new insights concern not only the awareness of some IBS-like disorders still underestimated today, but also the way in which they often overlap, making the clinical picture more complex to interpret. They are especially summarized in Figure 4, 5 and new Figure 6 which has been added, as also requested by reviewer 2. We also inserted strengthening specification in the conclusion section, so as to make the “take-home” message is more effective and the title is more appropriate.

  1. I suggest adding a paragraph in "2. Celiac Disease and Gluten-Related Disorders" that provides an overview of the necessity of a gluten-free diet in these conditions.

Thank you for your suggestion. Appropriate changes have been made.

  1. Figure 1 is included in the document but lacks mention in the main text.

Thank you for your suggestion. Appropriate changes have been made.

  1. Figure 5 shows potential clinical overlaps between IBS and other IBS-like disorders. Yet, ATIs, FODMAPs, and SIBO are listed as factors rather than diseases. Please correct Figure 5 accordingly.

Thank you for your suggestion. Appropriate specification has been made in the figure caption.

  1. It is advised to avoid using single-sentence paragraphs, such as those found in lines 188-189.

Thank you for your suggestion. Appropriate changes have been made.

Reviewer 2 Report

Comments and Suggestions for Authors

I've read with great interest the paper "Gluten-Free Diet for Fashion or Necessity: A Double-Edged Sword?", which refers to the indications of a gluten free diet (GFD) in gluten-related disorders in contrast with promoting the diet as a fashion statement, with its negative consequences.

The paper is of great interest to the current literature, and focuses on the deleterious effects of the GFD and the consequent increase in high-FODMAP foods, potential triggering of nickel allergic contact mucositis and exacerbation of histamine intolerance.

Figures are very representative and valuable for the manuscript.

Some comments for the authors

- it would be recommended to add in the introduction the benefits of GFD in CD with regard to impact on nutritional status, malignancy risk

- what's the clinical approach the authors propose in patients on a GFD who develop digestive symptoms? (patch tests, histamine testing); providing an algorithm to investigate these patients would be of added value.   

- there is a lot of emerging data regarding the metabolic risk in CD patients under GFD, expanding this topic might contribute to balancing the decision of a GFD in non-CD individuals

Author Response

REVIEWER 2 COMMENTS and ANSWERS

1 - it would be recommended to add in the introduction the benefits of GFD in CD with regard to impact on nutritional status, malignancy risk.

Thank you for your suggestion. Appropriate changes have been made. We have highlighted all the possible benefits of a GFD from multiple points of view (intestinal absorption, nutrition, signs and symptoms, avoidance of possible comorbidities and complications).

2 - what's the clinical approach the authors propose in patients on a GFD who develop digestive symptoms? (patch tests, histamine testing); providing an algorithm to investigate these patients would be of added value.   

Thank you for the valuable suggestion. We have inserted in the conclusion section an additional figure (figure 6) that proposes a possible diagnostic and therapeutic algorithm.

3 - there is a lot of emerging data regarding the metabolic risk in CD patients under GFD, expanding this topic might contribute to balancing the decision of a GFD in non-CD individuals.

Thanks for your suggestion. We have stressed this thought in the discussion section and also added a bibliographic source about it (please find the new refence 30).

Reviewer 3 Report

Comments and Suggestions for Authors

The topic taken up is important for medical and social reason. The purpose of the work and its implementation

does  not raise  any significant notes.

The work needs minor correction.  I suggest  that the final conclusion in the abstract should be more responsive to the question in the title of the paper.

Author Response

REVIEWER 3 COMMENTS and ANSWERS

The topic taken up is important for medical and social reason. The purpose of the work and its implementation does not raise any significant notes.The work needs minor correction.  I suggest  that the final conclusion in the abstract should be more responsive to the question in the title of the paper.

Thanks for your suggestion. We included an explicit answer to the title question at the end of the abstract.